# Impact of the Voltage-Gated Calcium Channel Antagonist Nimodipine on the Development of Oligodendrocyte Precursor Cells

**DOI:** 10.3390/ijms24043716

**Published:** 2023-02-13

**Authors:** Michael Enders, Alicia Weier, Rittika Chunder, Young An, Franziska Bremm, Andreas Feigenspan, Christian Buettner, Arif Bülent Ekici, Enrico Mingardo, Benjamin Odermatt, Stefanie Kuerten

**Affiliations:** 1Institute of Neuroanatomy, Medical Faculty, University of Bonn, 53115 Bonn, Germany; 2Institute of Anatomy and Cell Biology, University of Erlangen-Nuremberg, 91054 Erlangen, Germany; 3Department of Biology, Animal Physiology, University of Erlangen-Nuremberg, 91058 Erlangen, Germany; 4Institute of Human Genetics, University Hospital Erlangen, University of Erlangen-Nuremberg, 91054 Erlangen, Germany; 5Institute of Anatomy and Cell Biology, Medical Faculty, University of Bonn, 53115 Bonn, Germany

**Keywords:** dihydropyridines, nimodipine, MS, myelination, neuroprotection, Oli-Neu, OPC, zebrafish

## Abstract

Multiple sclerosis (MS) is an inflammatory demyelinating disease of the central nervous system (CNS). While most of the current treatment strategies focus on immune cell regulation, except for the drug siponimod, there is no therapeutic intervention that primarily aims at neuroprotection and remyelination. Recently, nimodipine showed a beneficial and remyelinating effect in experimental autoimmune encephalomyelitis (EAE), a mouse model of MS. Nimodipine also positively affected astrocytes, neurons, and mature oligodendrocytes. Here we investigated the effects of nimodipine, an L-type voltage-gated calcium channel antagonist, on the expression profile of myelin genes and proteins in the oligodendrocyte precursor cell (OPC) line Oli-Neu and in primary OPCs. Our data indicate that nimodipine does not have any effect on myelin-related gene and protein expression. Furthermore, nimodipine treatment did not result in any morphological changes in these cells. However, RNA sequencing and bioinformatic analyses identified potential micro (mi)RNA that could support myelination after nimodipine treatment compared to a dimethyl sulfoxide (DMSO) control. Additionally, we treated zebrafish with nimodipine and observed a significant increase in the number of mature oligodendrocytes (* *p*
≤ 0.05). Taken together, nimodipine seems to have different positive effects on OPCs and mature oligodendrocytes.

## 1. Introduction

Multiple sclerosis (MS) is a chronic neuroinflammatory disease thought to be caused and sustained by autoimmune responses against central nervous system (CNS) antigens, in particular, myelin antigens. One of the characteristic pathological hallmarks of MS is the presence of demyelinated focal lesions that develop on the background of inflammatory processes [1]. Inflammation in the MS-affected brain is associated with cellular and humoral components of both the innate and adaptive immune systems. On the one hand, diffuse infiltrates in the parenchyma of the CNS, perivascular cuffs, and meningeal cuffs consist of CD3^+^ T cells, CD20^+^ B cells, and plasma cells. On the other hand, activated microglia and macrophages mediate active tissue damage [2]. The inflammatory response is also driven by soluble factors, such as proinflammatory cytokines, chemokines, components of the complement system, and antibodies [3]. 

Given the consensus that inflammation drives demyelination in MS patients, most of the current therapies rely on immune-modulating strategies that either dampen the overall activity of the immune system or block the entry of peripheral immune cells into the CNS [4]. However, these therapies often fail in patients with progressive MS [5], where chronic demyelination and neuroaxonal loss are the key to irreversible neurological disability [6,7]. Although natural partial remyelination can be detected in a small percentage of patients post-mortem, it is neither efficient nor sufficient to halt axonal atrophy and loss [8]. Therefore, therapeutic strategies that promote neuroprotection and enhance remyelination are considered crucial for lesion repair and the restoration of neuronal activity [6], especially in the progressive phase of the disease. 

Several targets that therapeutically facilitate endogenous remyelination have been identified in preclinical studies [9]. For example, in vitro and in vivo studies have demonstrated that opicinumab, a monoclonal antibody against leucine-rich repeat and immunoglobin-like domain-containing protein (LINGO)-1, a negative regulator of myelination, enhanced oligodendrocyte differentiation and myelination [10,11]. Similarly, blocking neurite outgrowth factor (NOGO)-A has been shown to enhance neuronal plasticity and remyelination in a mouse model of MS [12,13]. However, neither opicinumab nor the anti-NOGO-A antibody could be translated into clinical trials. 

Another promising therapeutic target is nimodipine, which belongs to the group of dihydropyridines, a group of drugs capable of blocking the L-type voltage-gated calcium channels (VGCCs) Ca_v_1.1 to Ca_v_1.4 [14]. Nimodipine is typically used to treat vasospasms after subarachnoid hemorrhage [15], where its proposed effect is the relaxation of cerebral vascular smooth muscles [16]. Interestingly, more recent research has revealed that nimodipine has other modes of action. Not only did nimodipine increase the memory function of old [17] and young rats [18], but it also improved cognitive dysfunction after cerebral ischemia in a rat model, indicating a more global effect on the CNS [19]. Additionally, nimodipine showed a beneficial and remyelinating effect in experimental autoimmune encephalomyelitis (EAE), a mouse model of MS [20,21]. Nimodipine also positively affected astrocytes, neurons, and mature oligodendrocytes in culture [22,23,24,25]. 

In this current study, we examined the effect of nimodipine on oligodendrocyte precursor cells (OPCs), which are a natural reservoir of oligodendrocytes. Our data indicate that nimodipine does not impact the expression of myelin proteins; however, it could influence the expression of microRNA (miRNA), which supports myelin formation. 

## 2. Results

### 2.1. Oli-Neu Cells Share Common Characteristics with OPCs and Express Functional Ca_v_1.2

In the first set of experiments, we characterized the murine OPC cell line, Oli-Neu, which has previously been shown to have OPC properties [26]. Oli-Neu cells were stimulated with 1 µM PD 174265, which is a selective inhibitor of epidermal growth factor receptor, as described by Simon et al. [27]. Following stimulation, the cells started to differentiate into oligodendrocytes with high expression of myelin proteins, such as myelin basic protein (MBP) and myelin-associated glycoprotein (MAG) (Figure 1A). Furthermore, end-point PCR revealed a constant expression of MBP and proteolipid protein (PLP) and an increasing expression of MAG and myelin oligodendrocyte glycoprotein (MOG) during differentiation (Figure 1B). Given that Ca_v_1.2 and Ca_v_1.3 are the two VGCCs expressed by oligodendrocytes and OPCs [28], we also demonstrated the expression of Ca_v_1.2 but not Ca_V_1.1, Ca_v_1.3, and Ca_V_1.4 at the mRNA level by differentiated Oli-Neu cells. Patch-clamp experiments were conducted to demonstrate that the detected Ca_V_1.2 channel was physiologically active (Figure 1C). Further, those channels could be blocked by nimodipine and activated by BayK8644, a VGCC agonist (Figure 1C). 

### 2.2. Nimodipine Does Not Have an Effect on Myelin Gene Expression in Oli-Neu Cells

To investigate the effect of nimodipine on myelination, terminally-differentiated Oli-Neu cells were stimulated either with 1 µM or 10 µM nimodipine or dimethyl sulfoxide (DMSO) for 1, 4, and 6 days in culture medium. The experimental approach is comparable to a study already published by our group [25], where similar dose-titrated amounts of nimodipine were added to the oligodendroglial cell line, OLN-93, under differentiating conditions. Here, we measured the mean fluorescence intensity (MFI) and the extent of branching in Oli-Neu cells. MFI was defined as the ratio of the total fluorescence of a single image divided by the number of 4′,6-diamidino-2-phenylindole (DAPI)-positive cells. Branching was defined as the area occupied by a cell that stained positive for MBP (Figure 2A) or MAG (Figure 2B) after treatment with nimodipine versus DMSO. Quantification using artificial intelligence (AI)-based software revealed that the stimulation of Oli-Neu cells with PD 174265 increased the branching as well as the MFI for MBP and MAG (Figure 2C,D,F,G), indicating that the cells displayed a more mature status after one day in the culture medium. No changes in the expression of different myelin proteins were detectable when nimodipine was added to Oli-Neu cells, irrespective of the number of days during which the cells were kept in the culture medium (Figure 2C,D,F,G). Furthermore, using quantitative polymerase chain reaction (qPCR), no differences in the expression of myelin genes between nimodipine-treated cells and DMSO-treated cells were detected (Figure 2E,H). These results demonstrate that nimodipine did not significantly increase myelin-related gene and protein expression in the Oli-Neu cell line. 

### 2.3. Nimodipine Does Not Have Any Effect on Myelin Protein Synthesis in Primary OPCs from Neonatal Mice

To further confirm that nimodipine does not have any major effect on myelin-related genes in OPCs, we isolated primary OPCs from neonatal mice and characterized them following stimulation with platelet-derived growth factor (PDGF) and fibroblast growth factor 2 (FGF-2). Cells expressed Olig2 and SOX10 after one day in culture, indicating their differentiation into oligodendrocytes (Figure 3A). Additionally, the myelin proteins, MBP and MAG, were expressed, confirming the branching of the OPCs (Figure 3A). As shown in Figure 3B, the expression of myelin-related genes by the primary OPCs was comparable to that of the Oli-Neu cells (Figure 1). In addition to Ca_v_1.2, primary OPCs also expressed Ca_v_1.3 (Figure 3B). Patch-clamp analyses demonstrated that both Ca_v_1.2 and Ca_v_1.3 were physiologically active and could be blocked by nimodipine. Furthermore, BayK8644 was able to enhance the activation of VGCCs, as shown in Figure 3C. When treating primary OPCs with nimodipine versus DMSO, we did not detect any changes in the expression of MBP (Figure 4A–D) and MAG (Figure 4E–H) neither at the protein level nor at the mRNA level, which was similar to our results using the Oli-Neu cells (Figure 2). 

### 2.4. Effect of Nimodipine on the mRNA Level in OPCs

Given that both OPCs and Oli-Neu cells behaved similarly following nimodipine treatment in terms of their expression of myelin-related genes and VGCCs, we used Oli-Neu cells to further study the effect of nimodipine at the transcriptomic level. RNA sequencing (Figure 5) revealed an upregulation of genes, such as NADH-cytochrome b5 reductase 1 (*Cyb5r1*)*,* that support myelination [29], while others like metallothionein-1 (*Mt1)* were either downregulated or remained unchanged in the nimodipine-treated and DMSO-treated groups [30]. A summary of the results is provided in Table 1. Subsequently, further analyses were performed using the QIAGEN Ingenuity^®^ Pathway Analysis (IPA) to identify pathways that could be influenced by nimodipine. Top hits with a z-score of >+1 or <−1 are displayed in Table 2. Additionally, IPA revealed that several miRNAs were influenced by nimodipine. All of these miRNAs have been reported to be involved in the induction of differentiation and transformation of OPCs into mature oligodendrocytes. In MS patients, miRNA expression has been shown to be disturbed in the CNS gray matter [31,32], white matter [31,33], and peripheral blood [34]. A summary of the miRNAs of interest is provided in Table 3. 

### 2.5. Nimodipine Increases the Number of Oligodendrocytes, but Not OPCs, in an In Vivo Model

Finally, to confirm that nimodipine does not have any direct (re-) myelinating effect on OPCs, we used two zebrafish models in which green fluorescent protein (GFP) is expressed either under the Olig2 or the claudinK promoter. Three individual experiments were conducted with *n* = 8 fish per experiment. Cells were counted in the dorsal spinal cord after treatment of the fish with either 1 µM or 10 µM nimodipine or DMSO, respectively. Counting was performed using an AI-based software (Perkin Elmer) for Olig2 or by eye for claudinK. ClaudinK is a myelin protein expressed in zebrafish and is an analog of Claudin11, an essential component of CNS myelin in mammals [70,71]. As shown in Figure 6, the number of dorsal claudinK^+^ cells was increased compared to the control, indicating that there was an effect of nimodipine on mature oligodendrocytes (Figure 6C). On the contrary, there was no effect on dorsally migrated Olig2^+^ cells (Figure 6B). 

## 3. Discussion

This study aimed to investigate the effect of nimodipine on myelin gene and protein expression as well as on the differentiation and maturation of the Oli-Neu cell line and primary OPCs. Using different approaches, we were able to demonstrate that nimodipine did not enhance myelin production in our in vitro cell culture model, neither at the protein nor at the mRNA level. Furthermore, nimodipine did not induce any morphological alterations in the cells. However, bulk mRNA sequencing of Oli-Neu cells, followed by QIAGEN Ingenuity^®^ Pathway Analysis of the genes, revealed that nimodipine had the potential to modulate mRNA and miRNA expression patterns. 

In the first set of experiments, we confirmed the expression and physiological activity of L-type VGCCs in Oli-Neu cells and demonstrated that nimodipine, a canonical antagonist of Ca_V_1.2, was able to block the channel. However, nimodipine did not induce any upregulation or downregulation of myelin-related genes, which was further confirmed in primary OPCs. Using immunocytochemistry (ICC), we also analyzed Oli-Neu cells and OPCs for the expression of MBP, which is one of the most abundant myelin proteins. Additionally, MAG expression was measured at the protein level since it has been described as one of the terminal maturation proteins in oligodendrogenesis [72]. Again, there was no significant difference observed between nimodipine-treated and DMSO-treated cells. Subsequently, the transcriptome of the Oli-Neu cells was studied further using RNA-Seq. Several genes that are known to support myelination were differentially regulated in nimodipine-treated Oli-Neu cells. Among them was *Cntfr*, which is known to increase the expression of MOG [48]. Another example was *Gap43*, which negatively correlated with the expression of myelin in the gray matter [41] and was decreased after treatment with nimodipine. On the contrary, *Ddit3*, which was found to be upregulated during the demyelination [39,42], showed increased expression after treatment with nimodipine. Precisely what factors influence this contradictory effect of nimodipine on Oli-Neu cells requires further clarification. 

IPA-based bioinformatical analysis was performed to identify potential pathways that could be influenced by nimodipine [73], which resulted in the detection of several miRNAs that were affected (Table 3). miRNAs that belong to small non-coding RNAs (ncRNAs) have the ability to control gene expression in a sequence-specific manner [74] and are known to modulate >50% of protein-coding genes at the translational level [75]. The effect of nimodipine on repairing ischemic neuronal injury as well as inhibiting Tau phosphorylation in tauopathies through a miRNA-dependent axis has been previously discussed [76,77]. In our study we noticed, for example, that the levels of miR-219-2-3p and miR-219b-3p, two well-described miRNAs involved in (re-) myelination, were increased in Oli-Neu cells after nimodipine treatment [56,57,58,59]. These miRNAs have not only been identified as regulators of OPC maturation and myelin repair but have also been demonstrated to be biomarkers in MS patients [56,57,58,59]. 

Therefore, it is conceivable that nimodipine has the ability to support (re-)myelination by influencing specific miRNA and mRNA expression patterns. Future experiments should focus on identifying the precise interacting mRNA target(s) of miRNAs affected by nimodipine by performing relevant knock-in and knock-out studies both in vitro and in vivo. 

Finally, we confirmed our in vitro data with in vivo studies by treating zebrafish with either nimodipine or DMSO. The zebrafish model was chosen because of the rapid ability of zebrafish oligodendrocytes to myelinate [78]. The experiments revealed that only mature claudinK^+^ oligodendrocytes increased in number, in contrast with the total number of Olig2^+^ oligodendrocytes that are said to be mainly of a more immature phenotype [70]. 

In line with the data presented here, previous studies have demonstrated that nimodipine treatment increased the number of oligodendrocytes in the spinal cord in EAE, a mouse model of MS [20,21]. Additionally, increased remyelination was observed both in EAE [20,21] as well as in the cuprizone model of demyelination [24]. Furthermore, nimodipine exerted a beneficial effect on the rat oligodendrocyte cell line OLN-93 by upregulating myelin-related genes [25]. In our previous studies, the observed effects were independent of the interaction between nimodipine and the L-type VGCCs Ca_v_1.2 and Ca_v_1.3 [20,25]. Therefore, taken together, we suggest that not only does nimodipine have alternative interacting partners that are not limited to the L-type VGCCs Ca_v_1.2 and Ca_v_ 1.3, but the differential effects of nimodipine could be linked to specific stages of oligodendrocyte maturation [25]. Other alternative Ca^2+^ channels that nimodipine interacts with could be P2X_7_ channels which are also discussed as receptors for targeting neuroprotection [79,80]. 

In conclusion, nimodipine is a well-characterized and well-tolerated drug that has been used in clinical practice for more than four decades [15,81,82]. Most of the currently available therapies for the demyelinating disease MS are based on immune modulation and immune suppression, while remyelinating treatment is still missing [83]. MS, one of the most prevalent neurological diseases in young adults in the Western world, causes a high socio-economic burden [77]. The introduction of a neuroprotective drug would elevate MS treatment to a new level. Future studies will have to demonstrate if nimodipine has the potential to become such a drug. Although beneficial effects of nimodipine have been shown on remyelination both in vivo [20,21] and in vitro, in addition to the effects on astrocytes, microglia, and neurons [20,22,23,25,84], these effects could not be explained by a VGCC-dependent mode of action in all cases [22,25,84]. Therefore, further investigation is necessary to identify potential interaction partners of nimodipine at a cellular level, which would clarify its role and potential in (re-) myelination.

## 4. Materials and Methods

### 4.1. Nimodipine and BayK8644

Nimodipine and BayK8644 were provided (batch: BXR4H3P, research grade) by Bayer AG (Leverkusen, Germany) on the basis of a material transfer agreement. Both substances were dissolved in DMSO (Thermo Fisher Scientific, Waltham, MA, USA) to obtain a 10 mM stock solution. We used a dose of 1 µM or 10 µM for in vitro experiments following recommendations of previously published studies [85,86,87,88,89,90] and our own experience [20].

### 4.2. Oli-Neu Cell Line

The mouse OPC cell line Oli-Neu was provided by Jaqueline Trotter (University of Mainz, Mainz, Germany). Cells were grown in T75 cell culture flasks (Thermo Fisher Scientific) using the SATO medium containing Dulbecco’s modified Eagle’s medium (DMEM; Thermo Fisher Scientific) supplemented with 2% heat-inactivated horse serum (Themo Fisher Scientific), 5 µg/mL insulin (Merck, Darmstadt, Germany), 1% penicillin/streptomycin (10,000 U/mL) (Thermo Fisher Scientific), 1% N2 supplement (Thermo Fisher Scientific), 5 ng/mL sodium selenite (Merck), 25 µg/mL gentamicin (Merck, Darmstadt, Germany), 400 nM 3,3′,5-triiodo-L-thyronine (T3) (Merck), and 520 nM L-thyroxine (T4) (Merck). Cells were maintained at 37 °C and 5% CO_2_ and passaged when a confluency of 70–80% was reached. All cell culture materials were coated overnight at 37 °C with 0.0001% poly-ornithine (PORN) (Merck) solution. 

### 4.3. ICC

In preparation for immunocytochemical staining, 12-mm diameter coverslips (Carl Roth, Karlsruhe, Germany) were sterilized by heating at 60 °C overnight. For characterization, 2.5 × 10^4^ cells were seeded per well of a 24-well plate containing one coverslip each and incubated in the SATO medium for 48 h. Oli-Neu cells were stimulated with 1 µM PD 174265 [27] dissolved in DMSO, and OPCs were treated with 10 ng/mL PDGF-AA and 10 ng/mL FGF-2. For immunofluorescence analyses, cells were treated as above and additionally received 1 µM nimodipine, 10 µM nimodipine, or DMSO (Thermo Fisher Scientific), the latter as a vehicle. 

Cells were analyzed after 1, 4, and 6 days (Oli-Neu) and 1, 2, 4, and 6 days (OPCs), after which the medium was removed, and the cells were washed with PBS and fixed in ice-cold 4% (*v/v*) paraformaldehyde (Carl Roth) in PBS for 15 min. After the removal of paraformaldehyde, cells were washed three times with cold PBS and stored in PBS at 4 °C. For staining, cells were permeabilized using 0.1 M Triton X-100 (Carl Roth) in PBS at room temperature for 10 min, washed three times with PBS, and blocked with 10% BSA (Carl Roth) in PBS for 1 h at room temperature. After blocking, the following primary antibodies were diluted in 1% BSA in PBS: anti-MBP (Abcam, Cambridge, UK) (1:1000), anti-PLP (Novus, Abington, UK) (1:200), anti-Olig2 (Abcam) (1:250), anti-SOX10 (Abcam) (1:500), anti-MAG (Abcam) (1:100), and anti-MOG (Santa Cruz, Dallas, TX, USA) (1:200). Primary antibodies were incubated at 4 °C overnight. Cells were washed three times with PBS and incubated with secondary antibodies in 1% BSA in PBS using the following dilutions: goat anti-chicken Cy3 (Dianova, Hamburg, Germany) (1:500), goat anti-rabbit AlexaFluor^TM^ 488 (Dianova) (1:500), and goat anti-mouse AlexaFluor^TM^ 488 (Dianova) (1:400) for 1 h at room temperature. Subsequently, the cells were washed twice with PBS and once with ddH_2_O before coverslipping with fluoroshield + DAPI (Abcam). Images were acquired using a Leica DMi8 microscope (Leica, Wetzlar, Germany) at 40× magnification. Intercomputational clearing using the THUNDER^TM^ software was applied to the obtained images. For analyses, ten random images of one coverslip were taken. The fluorescence intensity and the total occupied area of the cells were measured using the NIS Elements^TM^ software (Nikon, Chiyoda, Japan) and divided by the number of visible cells to ensure a cell-based analysis.

### 4.4. RNA Isolation and PCR

Oli-Neu cells were differentiated using PD 174265, while primary OPCs were isolated using PDGF-AA (Miltenyi Biotech, Bergisch Gladbach, Germany) and FGF-2 (Miltenyi Biotech). For RNA isolation, the PureLink^TM^ Mini Kit with PureLink^TM^ DNase set (Thermo Fisher Scientific) was used per the manufacturer’s suggestions. The quantity and quality of the RNA were checked with a photometer. The RNA was reverse transcribed to complementary DNA (cDNA) using a reverse transcription kit (Thermo Fisher Scientific). 

For PCR, samples were amplified using a PCR master mix (Genaxxon, Ulm, Germany) with primers, as shown in Table 4. Gel electrophoresis was performed with a 2% agarose gel (Genaxxon) supplemented with GelRed^®^ DNA dye (Genaxxon). A DNA ladder of the size range 100 bp to 1.5 kbp (Genaxxon) was used to determine the size of the DNA fragments. Images were obtained using the iBright CL1500 Imaging System (Thermo Fisher Scientific). 

### 4.5. Patch-Clamp Analysis

Patch-clamp recordings were performed as described previously [91]. Briefly, coverslips with adherent oligodendrocytes were transferred to the recording chamber of an upright microscope (Zeiss, Jena, Germany) with a solution containing 137 mmol/L of NaCl, 5.4 mmol/L of KCl, 1.8 mmol/L of CaCl_2_, 1 mmol/L of MgCl_2_, 5 mmol/L of N-2-hydroxyethylpiperazine-N′-2-ethanesulfonic acid (HEPES), and 10 mmol/L of glucose (pH 7.4). The patch pipette solution contained 120 mmol/L of K-D-gluconate, 20 mmol/L of KCl, 1 CaCl_2_, 2 mmol/L of MgCl_2_, 11 mmol/L of 1,2-bis(o-aminophenoxy)ethane-N,N,N′,N′-tetraacetic acid (EGTA), and 10 mmol/L of HEPES (pH 7.2).

All experiments were carried out at room temperature (22 °C–23 °C). Voltage-gated currents were recorded with an EPC10 patch clamp amplifier (Heka Elektronik, Lambrecht, Germany) and low-pass filtered at 2.9 kHz using a built-in Bessel filter and digitized at 20 kHz with Patchmaster software (Heka Elektronik). Patch electrodes were pulled from borosilicate glass (Hilgenberg, Malsfeld, Germany) to a final resistance of 3–5 MΩ. Electrode tips were coated with Sylgard 184 (Dow Corning, Midland, MI, USA), and their series resistance (5–10 MΩ) was compensated up to 80%. Current traces were analyzed and plotted with OriginPro (OriginLab Corporation, Northampton, MA, USA) and Matlab (Mathworks, Natick, MA, USA).

### 4.6. Mice 

Male and female C57BL/6J mice (Charles River, Sulzfeld, Germany) were maintained under specific pathogen-free conditions at the Franz-Penzoldt-Zentrum (Palmsanlage 5, 91054 Erlangen, FAU Erlangen-Nürnberg, Germany) or at the House of Experimental Therapy (Nussallee 11, 53115 Bonn, University of Bonn, Germany). All animals had free access to a standard rodent diet (Ssniff, Soest, Germany) and autoclaved water. Room temperature was maintained between 20 °C–24 °C at 45%–65% humidity. Twelve-hour light/dark periods were ensured. All animal experiments complied with the German Law on the Protection of Animals (§4 and §7), the European Union directive 2010/63/EU, and the European Union regulation (EU) 2019/1010, the “Principles of Laboratory Animal Care” (NIH publication no. 86–23, revised 1985), as well as the “Animal Research: Reporting of In Vivo Experiments” (ARRIVE) guidelines [92]. 

### 4.7. OPC Isolation and Culture

Mice from postnatal day 1 (P1) to P3 were sacrificed by decapitation. The brain was dissected and stored in DMEM before it was transferred to C-Tubes (Miltenyi Biotec). For dissociation, the multi-tissue dissociation kit (Miltenyi Biotec) and the gentleMACS™ Octo Dissociator with Heaters (Miltenyi Biotec) with the program NKDT1 were used per the manufacturers’ protocols. Cell suspensions were filtered through a 70 µm filter (Sarstedt, Nümbrecht, Germany) and diluted in PBS containing 0.5% BSA. For the isolation of OPCs, the CD140a (platelet-derived growth factor receptor (PDGFR)α) microbead kit was used (Miltenyi Biotec) in combination with the QuadroMACS™ Separator (Miltenyi Biotec) with LS Columns (Miltenyi Biotec). In brief, columns were activated with 0.5% BSA/PBS solution. After samples had been applied, columns were washed and eluted with the same solution. OPCs were cultured in DMEM/F12 (Thermo Fisher Scientific) containing 1% N2 supplement, 2% B27 supplement (Thermo Fisher Scientific), 1% penicillin/streptomycin, 10 ng/mL PDGF-AA (Miltenyi Biotec), and 10 ng/mL FGF-2 (Miltenyi Biotec). Cell culture flasks were coated with 0.0075 µg/mL poly-D-lysine (Merck), and 24-well plates were coated with 0.0001 PORN before the cells were seeded.

### 4.8. Quantitative PCR

For qPCR, day 0 (before the addition of nimodipine/control) was used for baseline measurements. Cells were treated at identical time points as in ICC experiments. Experiments were performed in technical triplicates using a LightCycler 96 (Roche, Basel, Switzerland). *Actb* was used as an endogenous control. Relative gene expression was assessed using the ΔΔCT method [93]. Primers for qPCR are listed in Table 5.

### 4.9. Bulk mRNA-Seq

Total RNA was extracted from Oli-Neu cells using the PureLink RNA Mini Kit (Thermo Fisher Scientific), followed by DNase treatment. The quality of the isolated RNA samples was determined using an Agilent 2100 Bioanalyzer equipped with an Agilent RNA 6000 Nano Kit and related software (Agilent, Santa Clara, CA, USA). Sequencing libraries were generated from 1 mg of high-quality RNA using the TruSeq Stranded mRNA Kit (Illumina, San Diego, CA, USA) per the manufacturer’s instructions. Libraries were sequenced at a single end on a HiSeq 2500 platform (Illumina) with 100 bp reads to a depth of at least 56.3 million reads. Reads were converted to a FASTQ format with bcl2fastq (version 2.17.1.4; Illumina). Sequences were minimally quality-trimmed and adapter-trimmed using cutadapt (version 1.18; https://doi.org/10.14806/ej.17.1.200 (accessed on 5 February 2023)) The pre-processed reads were mapped to the *Mus musculus* reference genome GRCm38.6 and Ensemble gene model 102, using spliced transcripts alignment to a reference (STAR) software (Version 2.7.8.a; https://github.com/alexdobin/STAR (accessed on 5 February 2023)) and quantified as reads per gene while excluding exons shared between more than one gene (featureCounts [Version 2.0.1]; http://subread.sourceforge.net/ (accessed on 5 February 2023)). Based on the read count per gene, differentially expressed genes were determined using the negative binomial model as implemented in the differential gene expression analysis package DESeq2 (version 1.30; http://www.bioconductor.org/packages/release/bioc/html/DESeq2.html (accessed on 5 February 2023); R version 4.0.2). Results from significance tests were corrected for multiple testing using the Benjamini–Hochberg method [94]. IPA (Qiagen) was used to investigate pathways and up- and downstream regulators. An adjusted *p*-value of ≤0.01 was used as a cut-off to identify significantly differentially expressed genes. Raw data were uploaded on GEO and can be found under the identification number GSE221939. 

### 4.10. Fish

*Danio renio* was maintained under defined conditions [95] in our zebrafish core facility (Nussallee 10, 53115 Bonn, University of Bonn) according to the German law on the Protection of Animals (§11), the European Union directive 2010/63/EU, and the European Union regulation (EU) 2019/1010, the “Principles of Laboratory Animal Care” (NIH publication no. 86-23, revised 1985), as well as the “Animal Research: Reporting of In Vivo Experiments”(ARRIVE) guidelines [92]. All experiments were performed before the zebrafish larvae reached an age of 5 days after fertilization. 

A 14-h light period and a 10-h dark period were ensured. All animals were fed two times a day with artemia salinae and dry pellets (AquaPro2000, Bückenburg, Germany). *Tg(Olig2:eGFP)* [96] and *Tg(claudinK:eGFP)* [70] were paired as previously described [95]. After 30 min, eggs were collected and maintained in Danieau water at 28 °C. Methylene blue was used to avoid the growth of fungi. The pigmentation of fish was prevented using 0.003% phenylthiocarbamide (PTU) in Danieau water after the first 24 h. In addition, fish were decoronated and treated with 1 µM nimodipine, 10 µM nimodipine, or DMSO as a control for 48 h between 24 and 72 h post-fertilization. Fish were anesthetized with ethyl 3-aminobenzoate methanesulfonate (MS222) (Sigma), placed in 96-well glass plates (Cellvis, Mountain View, CA, USA) individually, and imaged with an EnSight Multimode Plate Reader (PerkinElmer, WA, USA). Z-stacks consisting of *n* = 6 images with z = 50 µm/step size were acquired at a wavelength of 465 nm and 50 ms light exposure. Olig2^+^ cells were counted using PerkinElmer’s Kaleido software (PerkinElmer) [55]. ClaudinK^+^ cells were counted manually using the ImageJ software. 

### 4.11. Statistics

For statistical analyses, Prism 9.3.1 was used (Graphpad, San Diego, CA, USA). Mean values ± standard deviations are indicated in all graphs. To test for normal distribution of data, the Shapiro–Wilk test was used. For the analysis of normally distributed data, ANOVA was used. The Kruskal–Wallis test was used when data were not normally distributed. A statistical significance level of 5% was chosen, and the *p*-values are displayed as * *p* ≤ 0.05, ** *p* ≤ 0.01, and *** *p* ≤ 0.001.

## Figures and Tables

**Figure 1 ijms-24-03716-f001:**
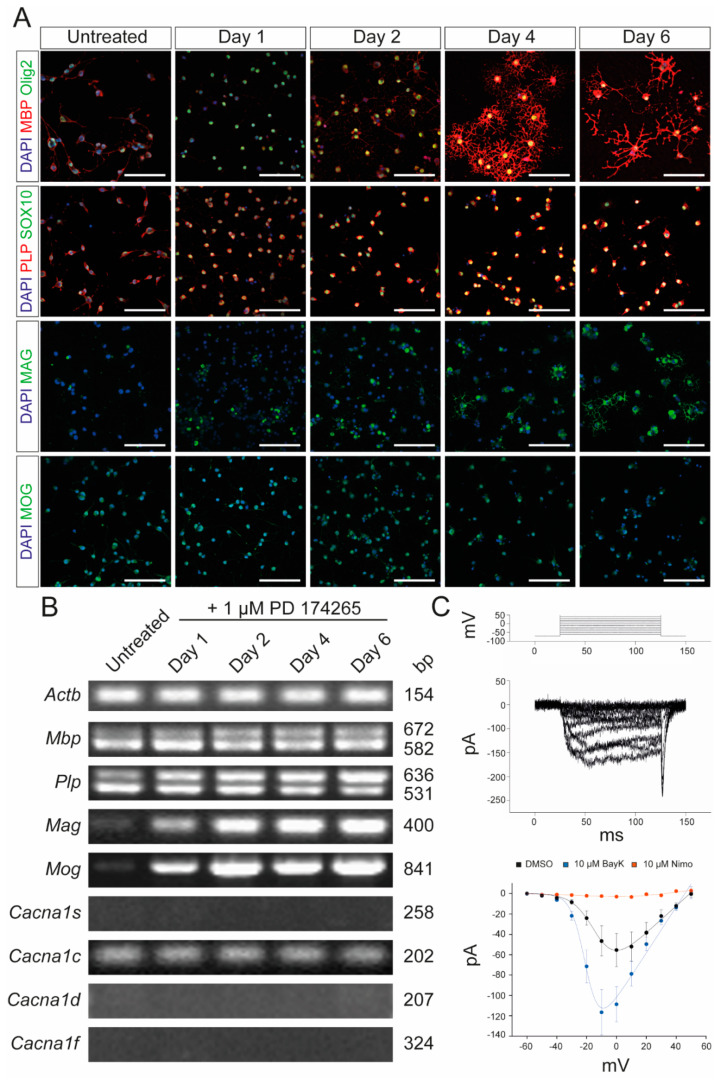
Characterization of Oli-Neu cells. (**A**) Immunocytochemical analysis of Oli-Neu cells with MBP (red)/Olig2 (green), PLP (red)/SOX10 (green), MAG (green), and MOG (green) after 1, 2, 4, and 6 days, respectively, following stimulation with 1 µM PD 174265 compared to untreated cells. Scale bars represent 100 µm. (**B**) End-point PCR showed increased expression of *Mag* and *Mog* after stimulation with 1 µM PD174265. Values on the right indicate the length of the PCR product in base pairs (bp). (**C**) Patch-clamp experiments showed the physiological presence of voltage-gated calcium channels that could be blocked by 10 µM nimodipine and activated by 10 µM BayK8644. DMSO was used as a control. The graph displays the data points of measured electric currents (Y-axis) against the applied voltages (X-axis). Error bars indicate standard deviations. Values are representative of *n* = 3 independent experiments and are presented as mean ± standard deviation. *Actb*: beta-actin; BayK: BayK8644; bp: base pairs; *Cacna1s*: voltage-gated L-type calcium channel 1.1; *Cacna1c*: voltage-gated L-type calcium channel 1.2; *Cacna1d*: voltage-gated L-type calcium channel 1.3; *Cacna1f*: voltage-gated L-type calcium channel 1.4; DAPI: 4′,6-diamidino-2-phenylindole; DMSO: dimethyl sulfoxide; MAG: myelin-associated glycoprotein; MBP: myelin basic protein; MOG: myelin oligodendrocyte glycoprotein; ms: milliseconds; mv: millivolt; pA: picoamperes; Nimo: nimodipine; Olig2: oligodendrocyte transcription factor 2; PCR: polymerase chain reaction; PLP: proteolipid protein; SOX10: SRY-box transcription factor 10.

**Figure 2 ijms-24-03716-f002:**
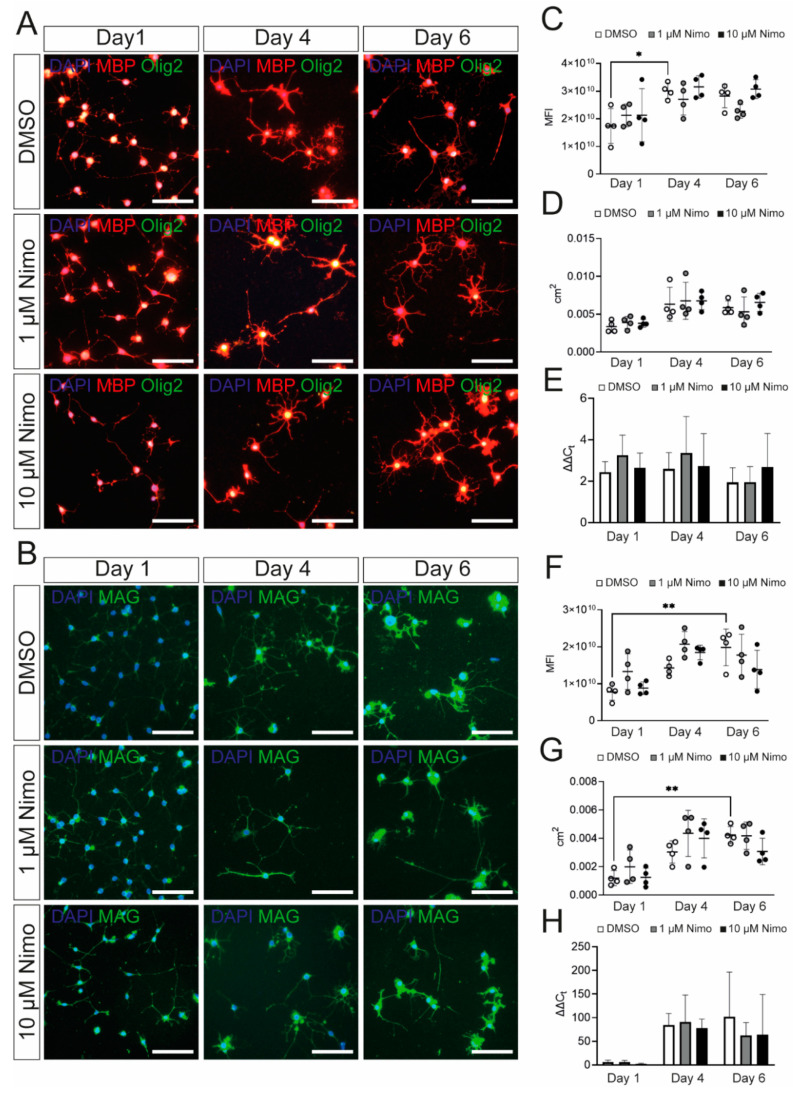
Oli-Neu cells show increased expression of myelin-related genes after PD 174265 treatment. (**A**,**B**) Staining of stimulated Oli-Neu cells treated with DMSO, 1 µM, or 10 µM nimodipine. Images show MBP (red)/Olig2 (green) (**A**) or MAG (**B**). Scale bars represent 100 µm. In *n* = 10 images, the MFI (**C**,**F**) and the area occupied by all cells per image in cm^2^ (**D**,**G**) were measured. (**E**,**H**) mRNA expression of *Mbp* (**E**) and *Mag* (**H**). Data are presented as mean value ± standard deviation. All experiments were performed as *n* = 4 independent experiments. Kruskal–Wallis test was performed to test for statistical significance. * *p*
≤ 0.05 ** *p*
≤ 0.01. cm^2^: square centimeters; DAPI: 4′,6-diamidino-2-phenylindole; DMSO: dimethyl sulfoxide; MBP: myelin basic protein; MFI: mean fluorescent intensity; Nimo: nimodipine.

**Figure 3 ijms-24-03716-f003:**
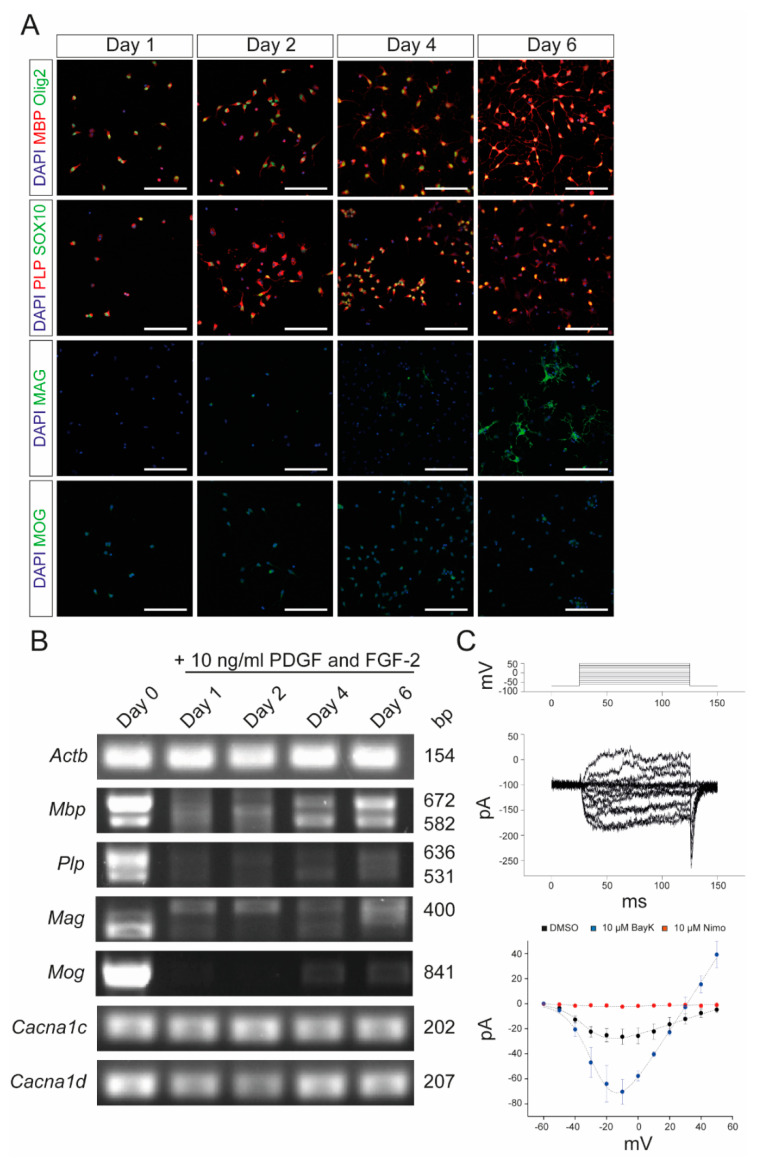
Characterization of primary OPCs after stimulation with PDGF and FGF-2. (**A**) Immunocytochemistry of OPCs with MBP (red)/Olig2 (green), PLP (red)/SOX10 (green), MAG (green), and MOG (green) after 1, 2, 4, and 6 days, respectively, following stimulation with 10 ng/mL PDGF and FGF-2. Scale bars represent 100 µm. (**B**) End-point PCR after stimulation with 10 ng/mL PDGF and FGF-2. Values on the right indicate the length of the PCR products in base pairs. (**C**) Patch-clamp experiments revealed the physiological presence of VGCCs, which could be blocked by 10 µM nimodipine and activated by 10 µM BayK8644. DMSO was used as control agent. Data are presented as mean ± standard deviation. All experiments were performed as *n* = 3 independent experiments. *Actb*: beta-actin; BayK: BayK8644; bp: base pairs; *Cacna1c*: voltage-gated L-type calcium channel 1.2; *Cacna1d*: voltage-gated L-type calcium channel 1.3; DAPI: 4′,6-diamidino-2-phenylindole; DMSO: dimethyl sulfoxide; FGF-2: fibroblast growth factor 2; *Mag*: myelin-associated glycoprotein; *Mbp*: myelin basic protein; *Mog*: myelin oligodendrocyte glycoprotein; ms: milliseconds; mV: millivolt; Nimo: nimodipine; pA: picoamperes; PDGF: platelet-derived growth factor; *Plp*: proteolipid protein; SOX10: SRY-box transcription factor 10; VGCC: voltage-gated L-type calcium channel.

**Figure 4 ijms-24-03716-f004:**
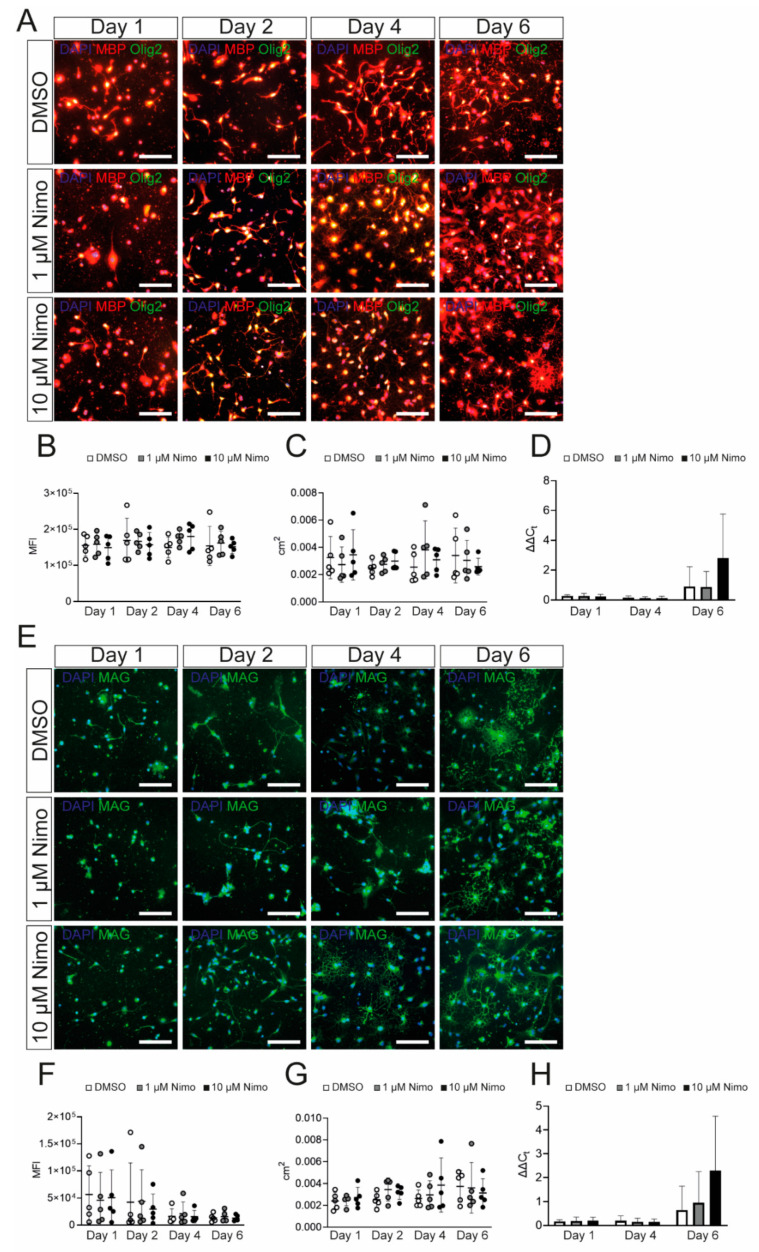
Nimodipine does not have an impact on myelin gene and protein expression by oligodendrocyte precursor cells. (**A**,**E**). Staining of oligodendrocyte precursor cells treated with DMSO, 1 µM nimodipine, or 10 µM nimodipine. Images show MBP (red)/Olig2 (green) (**A**) or MAG (**E**). DAPI indicates nuclear staining. Scale bars indicate 100 µm. The MFI (**B**,**F**) and the area occupied by all cells in each image (**C**,**G**) were measured in *n* = 10 images per condition. (**D**,**H**) mRNA expression of *Mbp* (**D**) and *Mag* (**H**). *n* = 5 independent experiments of every experiment were performed. Error bars indicate mean ± standard deviation. Kruskal–Wallis test was performed to test for statistical significance. cm^2^: square centimeters; DAPI: 4′,6-diamidino-2-phenylindole; DMSO: dimethyl sulfoxide; MAG: myelin-associated glycoprotein; MBP: myelin basic protein; MFI: mean fluorescent intensity; Nimo: nimodipine.

**Figure 5 ijms-24-03716-f005:**
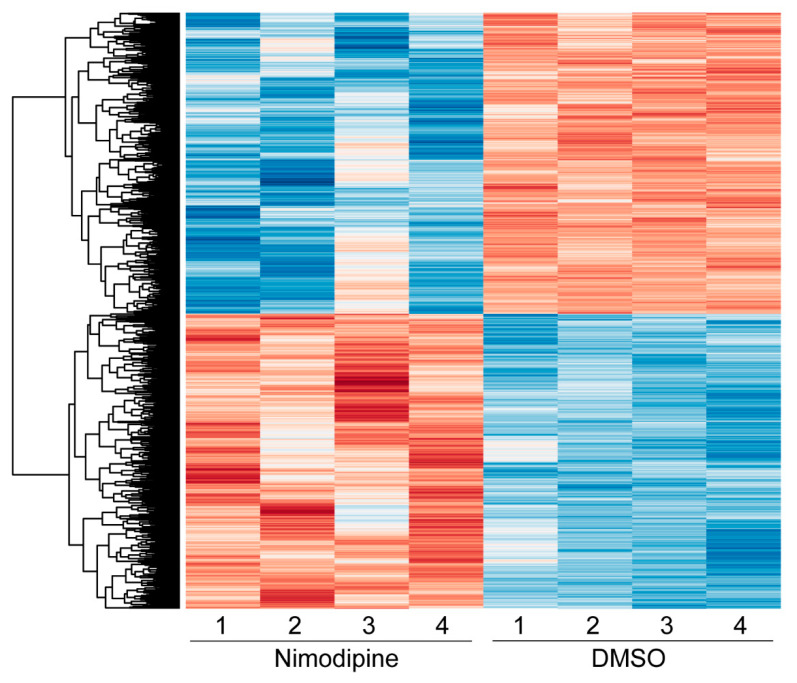
Heatmap showing the hierarchical clustering of all differentially expressed genes comparing nimodipine-treated and DMSO-treated Oli-Neu cells after 24 h. Red indicates upregulated genes, while blue indicates downregulated genes. The numbers 1, 2, 3, and 4 represent the individual experiments. Results from significance tests were corrected for multiple testing using the Benjamini–Hochberg method. DMSO: dimethyl sulfoxide.

**Figure 6 ijms-24-03716-f006:**
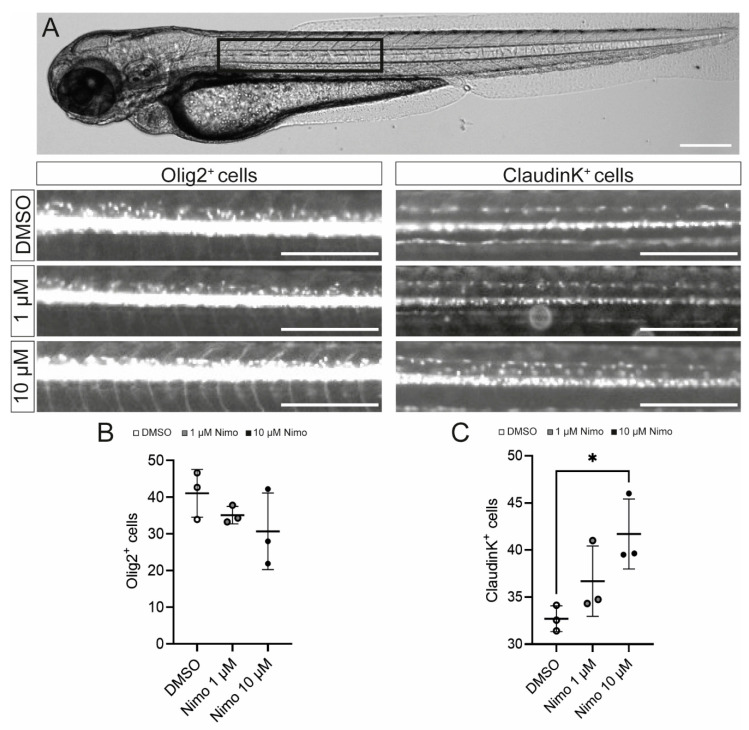
Nimodipine treatment increases the number of oligodendrocytes, but not oligodendrocyte precursor cells, in an in vivo zebrafish model. (**A**) Light microscopic image of a zebrafish. The black box highlights the area of interest that was used for quantitative cell analyses, as shown below. Scale bars indicate 250 µM. (**B**,**C**) Numbers of Olig2^+^ cells (**B**) and claudinK^+^ cells (**C**) in the area of interest. Ordinary one-way ANOVA was performed to test for statistical significance. * *p* < 0.05. A single data point represents a mean value of *n* = 8 zebrafish. Data are representative of *n* = 3 independent experiments. DMSO: dimethyl sulfoxide; Nimo: nimodipine.

**Table 1 ijms-24-03716-t001:** Changes in mRNA expression patterns after nimodipine treatment in Oli-Neu cells. Top-hit genes with *p <* 10^−15^ are shown.

Gene	Key Functions of the Gene with Its Relevance to This Study	Impact of Nimodipine	Sources
*Bhlhe40*	This growth factor increases the production of cytokines in T cells. The factor itself is regulated by the vitamin D receptor.	Increased expression	[35]
*Nr4a1*	NR41A is expressed in several cell types. Nr41a is known to prevent immune cell infiltration into the CNS.	Increased expression	[36,37]
*Mt1*	MT1 is capable of binding to free radicals and heavy metal ions and has neuroprotective potential in the CNS; *Mt1^−/−^* and *Mt2^−/−^* EAE mice display an increased number of CNS lesions.	Increased expression	[30,38,39]
*Cyb5r1*	CYB5R1 expression has been described to increase the brain volume in the early phases of development.	Increased expression	[29]
*Nr1d1*	This nuclear receptor displays higher expression in neuronal stem cells than fetal cells.	Increased expression	[40]
*Gap43*	The level of GAP43 correlates negatively with the formation of myelin in gray matter.	Decreased expression	[41]
*Ddit3*	DDIT3 is activated when demyelination occurs. In the cuprizone mouse model, *Ddit3^-/-^* knockout mice show reduced demyelination.	Increased expression	[42]
*Sesn2*	SESN2 has been described as a universal growth factor. In MS patients, serum levels of SESN2 are decreased.	Increased expression	[43,44]
*Gadd45b*	The DNA methylase, GADD45B, increases transcription and stimulates axonal regeneration.	Increased expression	[45]
*Serpini1*	The direct effect of SERPINI1 is unknown. It is speculated that there is a negative correlation between neuronal growth and SERPINI1 expression.	Decreased expression	[46]
*Pik3r3*	PIK3R3 regulates lipid metabolism and is part of a myelin synthesis pathway.	Increased expression	[47]
*Cntfr*	CNTFR controls the expression of MOG.	Increased expression	[48]

*Bhlhe40:* basic helix–loop–helix protein 40 class E; CNS: central nervous system; *Cntfr*: ciliary neurotrophic factor receptor; *Cyb5r1*: NADH-cytochrome b5 reductase 1; *Ddit3:* DNA damage-inducible transcript 3; EAE: experimental autoimmune encephalomyelitis; *Gadd45b*: growth arrest and DNA damage-inducible beta; *Gap43*: growth-associated protein 43; MOG: myelin oligodendrocyte glycoprotein; *Mt1:* metallothionein-1; *Nr1d1*: nuclear receptor subfamily 1 group D member 1; *Nr4a1*: nuclear receptor 4A1; *Pik3r3*: phosphatidylinositol 3-kinase regulatory subunit gamma; *Serpini1*: neuroserpin; *Sesn2*: sestrin-2.

**Table 2 ijms-24-03716-t002:** Changes in mRNA expression patterns after nimodipine treatment in Oli-Neu cells. Top hits, as identified by IPA analysis and with an activation z-score of >1 or <−1, are shown.

Gene	Key Functions of the Gene with Its Relevance to This Study.	Impact of Nimodipine	Sources
*Magi2*	MAGI2 has been described to be involved in synaptic transmission. MAGI2 expression was decreased in EAE.	Increased expression	[49]
*S100A2*	This calcium sensor is crucial for the organization of the cytoskeleton and the microtubule network in oligodendrocytes.	Increased expression	[50]
*Oxsr1*	OXSR1 is upregulated when embryonic stem cells change to OPCs.	Increased expression	[51]
*Tsc22d1*	TSC22D1 is a transcription factor that is upregulated in oligodendrocytes and downregulated in astrocytes.	Increased expression	[52]
*Cdc2*	CDC2, which regulates the cell cycle, is present in OPCs but absent in mature oligodendrocytes.	Increased expression	[53]
*Pfdn5*	PFDN5 is a molecular chaperone that stabilizes newly-synthesized proteins, allowing them to fold correctly. It is decreased in aging OPCs.	Decreased expression	[54]
*Mknk*	This kinase is responsible for lipid synthesis in oligodendrocytes.	Increased expression	[55]

*Cdc2*: cell division cycle protein 2 homolog; EAE: experimental autoimmune encephalomyelitis; *Magi2*: membrane-associated guanylate kinase inverted 2; *Mknk:* MAP kinase signal-interacting kinase; OPC: oligodendrocyte precursor cell; *Oxsr1:* Serine/threonine-protein kinase; *Pfdn5:* prefoldin subunit 5; *S100A2*: S100 calcium-binding protein A2; *Tdsc22d1:* TSC22 domain family protein 1.

**Table 3 ijms-24-03716-t003:** miRNAs affected by nimodipine treatment that is known to play a role in MS pathology.

miRNA	Expression Pattern	Known Interacting Partner	Impact of Nimodipine	Source
miR-106	Lower expression in blood samples and in gray matter lesions of MS patients	miR-106 is a negative regulator for the amyloid precursor protein	Increased expression	[34]
miR-219-2-3p	Lower expression in the cerebrospinal fluid of MS patients and in gray matter lesions	miR-219 is crucial for developing OPCs via, e.g., the suppression of caspase 3 expression	Increased expression	[56,57,58,59]
miR-219b-3p				
miR-20a	Lower expression in gray matter lesions in MS patients	miR-20a suppresses the expression of NF-κB and IL-17/IL-23	Increased expression	[60,61]
miR-34a-3p	Increased expression in patients with Alzheimer’s disease	miR-34a-3p acts as a target for CD47. CD47 is a “do not eat me” surface marker	Decreased expression	[62,63]
miR-34a-5p	Decreased expression in lesions of MS patients	miR-34a-5p reduces the expression of, e.g., sirtuin 1, which increases differentiation in neuronal cells	Increased expression	[62,63]
miR-23	Expression is reduced in gray matter lesions in MS patients	miR-23 is a negative regulator of lamin B1, which is essential for oligodendrogenesis	Increased Expression	[64,65]
miR-338	Decreased expression in white and gray matter lesions of MS patients	Interaction partner of miR-219; increases the expression of myelin	Increased expression	[58,59]
miR-7	Downregulation in white matter lesions of MS patients	miR-7 inhibits Wnt and activates Shh signaling, both of which increase myelination	Increased expression	[66]
miR-1275	Upregulation in white matter lesions of MS patents	Inhibition of miR-1275 increases the expression of claudin 11	Decreased expression	[67]
miR-122	Expression is increased in white matter lesions of MS patients	miR-122 expression increases the expression of proinflammatory type I interferons	Decreased expression	[68,69]

CD47: cluster of differentiation marker 47; IL: interleukin; miR: microRNA; MS: multiple sclerosis; Nf-κB: nuclear factor κ-light-chain-enhancer of activated B cells; OPC: oligodendrocyte precursor cell; Shh: sonic hedgehog; WNT: wingless-type**.**

**Table 4 ijms-24-03716-t004:** List of primers used for PCR.

Gene	Forward Primer	Reverse Primer	Annealing Temperature	Length (bp)
*Actb*	GGCTGTATTCCCCTCCATCG	CCAGTTGGTAACAATGCCATGT	55 °C	154
*Mbp*	CGGACCCAAGATGAAAACCC	AAAGGAAGCCTGGACCACACAG	56 °C	582, 672
*Plp*	GAAAAGCTAATTGAGACCTA	GAGCAGGGAAACTAGTGTGG	45 °C	531, 636
*Mag*	CTCTATGGCACCCAGAGCCT	TGTCCTTGGTGGGTCGTTTT	56 °C	355, 400
*Mog*	GACCTCAGCTTGGCCTGACCC	TGCTGGGCTCTCCTTCCGC	63 °C	841
*Cacna1s*	TGTGGTATGTCGTCACTTCCTCC	CGTCAATGATGCTGCCGATG	57 °C	258
*Cacna1c*	CAGCTCATGCCAACATGAAT	TGCTTCTTGGGTTTCCCATA	53 °C	202
*Cacna1d*	AATGGCACGGAATGTAGGAG	GACGAAAAATGAGCCAAGGA	52 °C	207
*Cacna1f*	GACGAATGCACAAGACATGC	CAAGCACAAGGTTGAGGACA	55 °C	324

*Actb*: beta-actin; bp: base pairs; *Cacna1c*: voltage-gated calcium channel 1.2; *Cacna1d*: voltage-gated calcium channel 1.3; *Cacna1f*: voltage-gated calcium channel 1.4; *Cacna1s*: voltage-gated calcium channel 1.1; *Mag*: myelin-associated glycoprotein; *Mbp*: myelin basic protein; *Mog*: myelin oligodendrocyte glycoprotein; *Plp*: proteolipid protein.

**Table 5 ijms-24-03716-t005:** List of primers used for qPCR.

Gene	Forward Primer	Reverse Primer	Ann. Temp.	Length (bp)
*Actb*	ACGATATCGCTGCGCTGG	AGCATCGTCGCCCGC	60 °C	71
*Mbp*	GGCACGCTTTCCAAAATCT	GGAGGGCTCTCAGCGTCT	60 °C	81
*Plp*	TCAGTCTATTGCCTTCCCTAGC	AGCATTCCATGGGAGAACAC	60 °C	91
*Mag*	CCAGTACACCTTCTCGGAGC	TCCGGCACCATACAACTGAC	60 °C	114
*Mog*	ATGCTTACATGGAGGTTGGGC	TTCCTCCAAGAAGCCCGAAG	60 °C	70

*Actb*: beta-actin; bp: base pairs; *Mag*: myelin-associated glycoprotein; *Mbp*: myelin basic protein; *Mog*: myelin oligodendrocyte glycoprotein; *Plp*: proteolipid protein.

## Data Availability

The data generated and analyzed during the current study are available from the corresponding author upon reasonable request.

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
