# Peer review of "Impact of the Voltage-Gated Calcium Channel Antagonist Nimodipine on the Development of Oligodendrocyte Precursor Cells"

_ijms, 2023, doi:10.3390/ijms24043716_

Round 1
Reviewer 1 Report
The work of Michael Enders and colleagues focuses on the problem of remyelination for the therapy of a known incurable disease such as multiple sclerosis. The authors demonstrated that the L-type voltage-gated calcium channel blocker, nimodipine, affects the expression of miRNAs involved in the remyelination process, but does not affect myelin synthesis and OPCs development. After reading the manuscript, I had several remarks regarding the rationale for some experimental approaches.
- Nimodipine, like other dihydropyridines, is able to block not only Cav1.2 and Cav1.3, but also other subtypes of L-type channels. The authors note this in the introduction. A similar effect exerts BayK8644 by activating all L-type calcium channels. The authors tested only the expression of mRNA Cav1.2 and Cav1.3 channels, citing that these channels are expressed by OPCs cells. However, this does not guarantee that Oli-Neu-Cells express only Cav1.2 and Cav1.3 channels. Therefore, it is necessary to prove the absence of expression of other L-type channels by at least PCR.
- What is the basis for choosing nimodipine 1 and 10 μM concentrations? For dihydropyridines, effective blocking of L-channels in the nanomolar range was shown, whereas high concentrations of these compounds have a side effect, including on neurons and astrocytes (10.1073/pnas.0936131100; 10.3390/ijms221910342).
Minor points
- In the case of demonstrating the significance of the differences in the figures, the figure legends should reflect which statistical test was used in one case or another.
- Please, explain what Vehicle means in plots with electrophysiological data.
Author Response
We thank the reviewers for taking their time to review the submitted manuscript and for providing instructive critiques and comments. In the following, we provide a point-by-point response to the comments:
Reviewer 1:
Nimodipine, like other dihydropyridines, is able to block not only Cav1.2 and Cav1.3, but also other subtypes of L-type channels. The authors note this in the introduction. A similar effect exerts BayK8644 by activating all L-type calcium channels. The authors tested only the expression of mRNA Cav1.2 and Cav1.3 channels, citing that these channels are expressed by OPCs cells. However, this does not guarantee that Oli-Neu-Cells express only Cav1.2 and Cav1.3 channels. Therefore, it is necessary to prove the absence of expression of other L-type channels by at least PCR.
We would like to thank the reviewer for this comment. We have performed additional PCR experiments as described in section 4.4. The primers have now been included in Table 3. The results demonstrate that there was no expression of CaV1.1 and CaV1.4 in Oli-Neu cells. Please refer to the updated Figure 1.
What is the basis for choosing nimodipine 1 and 10 μM concentrations? For dihydropyridines, effective blocking of L-channels in the nanomolar range was shown, whereas high concentrations of these compounds have a side effect, including on neurons and astrocytes (10.1073/pnas.0936131100; 10.3390/ijms221910342).
We would like to thank the reviewer for this remark. Indeed, nimodipine is capable of blocking voltage- gated calcium channels at a concentration of 200 nM (https://doi.org/10.1186/s12929-018-0447-z). We chose 1 μM and 10 μM as concentrations for in vitrouse for two reasons. On the one hand, we have previously successfully applied these concentrations for studies on microglia (please see reference 20 of the revised manuscript) and on the other hand, other groups have used similar concentrations. We have now elaborated it in the revised manuscript on page 16 and we have added the additional references 85-90.
In the case of demonstrating the significance of the differences in the figures, the figure legends should reflect which statistical test was used in one case or another.
We thank the reviewer for this comment. We have now added the statistical tests that were performed for the Figures 2, 4, 5 and 6 in their corresponding figure legends of the revised manuscript.
Please, explain what Vehicle means in plots with electrophysiological data.
We would like to apologize for the misunderstanding. We used DMSO as our vehicle. We have now replaced “vehicle” by “DMSO” in Figures 1 and 3.
Reviewer 2 Report
Dear Author/s,
This article investigates the effect of voltage-gated calcium channel antagonist ni-2 modipine on the development of oligodendrocyte precursor cells. The subject of the article is very original and the carefully written, evaluated orted by current scientific studies.
Specific comments:
- The “Abstract” section: Statistical significance value of the findings should be written.
- The "Materials and Methods" section: If any, information regarding the approval of the ethics committee in animal experiments should be added.
Doses of 1 μM nimodipine and 10 μM nimodipine were chosen based on which research. It must be specified.
Author Response
We thank the reviewers for taking their time to review the submitted manuscript and for providing instructive critiques and comments. In the following, we provide a point-by-point response to the comments:
Reviewer 2:
The “Abstract” section: Statistical significance value of the findings should be written.
We would like to thank the reviewer for this comment. We have now added the statistical significance values in the abstract.
The "Materials and Methods" section: If any, information regarding the approval of the ethics committee in animal experiments should be added.
We would like to thank the reviewer for this important comment. According to German and European law on the Protection of Animals, no ethics approval is needed when fish younger than 5 days after fertilization are used. This is now mentioned on page 20 of the revised manuscript. Experiments using primary OPCs complied with the German Law on the Protection of Animals (§4 and §7), and were in concordance with the European Union directive 2010/63/EU, the European Union regulation (EU) 2019/1010, the “Principles of Laboratory Animal Care” (NIH publication no. 86-23, revised 1985) as well as the Reporting Animal Research of in vivo Experiments (ARRIVE) guidelines. These informations are now mentioned on pages 18 and 19 of the revised manuscript.
Doses of 1 μM nimodipine and 10 μM nimodipine were chosen based on which research. It must be specified.
We would like to thank the reviewer for this remark. Indeed, nimodipine is capable of blocking voltage- gated calcium channels at a concentration of 200 nM (https://doi.org/10.1186/s12929-018-0447-z). We chose 1 μM and 10 μM as concentrations for in vitrouse for two reasons. On the one hand, we have previously successfully applied these concentrations for studies on microglia (please see reference 20 of the revised manuscript) and on the other hand, other groups have used similar concentrations. We have now elaborated it in the revised manuscript on page 16 and we have added the additional references 85-90.
Round 2
Reviewer 1 Report
The authors have addressed all my comments and made necessary corrections.